# Immune Checkpoint Inhibitors and the Kidney: A Focus on Diagnosis and Management for Personalised Medicine

**DOI:** 10.3390/cancers15061891

**Published:** 2023-03-21

**Authors:** Elisa Longhitano, Paola Muscolino, Claudia Lo Re, Serena Ausilia Ferrara, Valeria Cernaro, Guido Gembillo, Dalila Tessitore, Desirèe Speranza, Francesco Figura, Mariacarmela Santarpia, Nicola Silvestris, Domenico Santoro, Tindara Franchina

**Affiliations:** 1Unit of Nephrology and Dialysis, Department of Clinical and Experimental Medicine, A.O.U. “G. Martino”, University of Messina, 98125 Messina, Italy; 2Medical Oncology Unit, Department of Human Pathology “G. Barresi”, University of Messina, 98125 Messina, Italy

**Keywords:** immune checkpoint inhibitors, ICIs, irAE, CTLA-4, PD-1, AKI-ICI, cancer

## Abstract

**Simple Summary:**

Immune-oncology has revolutionized the natural history of many cancers in the past decade, becoming a new therapeutic weapon. The identification of immune “checkpoints” (PD-1/CTLA-4) whose blockade stimulates anti-tumor immunity has changed outcomes for many ptients. However, immune checkpoint inhibitors (ICIs) produce novel response patterns across cancer types and can cause inflammatory side-effects known as immune-related adverse events (irAEs). Renal damage from ICIs is an infrequent event, and early diagnosis plays a key role in its treatment, so it is crucial to recognize and treat it early to avoid toxicities of all grades, as well as hospitalizations.

**Abstract:**

Immunity plays a crucial role in fighting cancer, but tumours can evade the immune system and proliferate and metastasize. Enhancing immune responses is a new challenge in anticancer therapies. In this context, efficacy data are accumulating on immune checkpoint inhibitors and adjuvant therapies for various types of advanced-stage solid tumours. Unfortunately, immune-related adverse events are common. Although infrequent, renal toxicity may occur via several mechanisms and may require temporary or permanent drug suspension, renal biopsy, and/or immunosuppressive treatment. This short review aims to provide a practical approach to the multidisciplinary management of cancer patients with renal toxicity during treatment with immune checkpoint inhibitors.

## 1. Introduction

The ability of the immune system to recognise self-cells and specific “non-self” antigens underlies the body’s defence mechanisms [1,2]. In this context, many inhibitory signal transduction pathways maintain immunological tolerance and cellular homeostasis [3,4]. Their central role is to protect healthy tissues from damage by preventing the excessive activation of T-cells [3,4]. However, in the context of the tumour’s ability to get through the immune response, regulating T-cell activation can allow for tumour growth [5,6].

Understanding the mechanisms underlying T-cell activation has led to the development of new anticancer treatments; however, the latter are associated with immune-related adverse events (irAEs), which may also affect the kidney [7].

Here, we summarise the current knowledge on renal involvement during immune-checkpoint inhibitor (ICI) therapies and describe a multidisciplinary approach for providing the most appropriate personalised care.

## 2. Immune Checkpoints and Cancer

T-cell activation is at the heart of the immune response against cancer, which begins with the recognition of tumour antigens by antigen-presenting cells (APCs) [1,2,5,8,9,10]. Several signals (antigen presentation, co-stimulatory signals, and cytokine expression) are needed to activate T-cells effectively, and many others are indispensable for avoiding self-cell attack [10,11]. Cytotoxic T-lymphocyte antigen 4 (CTLA-4) in lymphatic tissues and programmed death protein 1 (PD-1) and programmed death ligands 1 and 2 (PD-L1 and PD-L2) in peripheral tissues are key players in protecting healthy cells [11].

CTLA-4 is expressed on activated T-cells; its main function is to down-regulate T-cell activation by counteracting the co-stimulatory signal delivered by CD28 [10,12]. CTLA-4 and CD28 share the same ligands—CD80 (B7.1) and CD86 (B7.2)—in APC cells [10,13]. CTLA-4 has a higher affinity for these ligands and can, therefore, successfully compete with CD28 for ligand binding—attenuating the T-cell response [10,13].

PD-1 is an inhibitory receptor expressed in T-cells, B cells, and natural killer cells; it interacts with PD-L1 and PD-L2—expressed mainly in inflammatory tissues and tumour microenvironments. PD-1 and PD-L1/PD-L2 binding inactivates T-cells [10].

## 3. ICI

New anticancer treatments can target the regulators of the immune response to increase T-cell activity against cancer [10,11]. In this context, ICIs—humanised monoclonal antibodies—have proven to be excellent treatment options, improving progression-free and overall survival from various skin, haematological, pulmonary, uro-genital, and gastrointestinal tumours [14,15].

Anti-CTLA-4 prevents CTLA-4–B7 interactions by promoting CD28-mediated co-stimulatory pathways and T-cell responses [10]. Ipilimumab, an anti-CTLA-4 antibody, was the first immunotherapy drug to demonstrate improved overall survival in patients with advanced melanoma in a Phase 3 study [14].

Similarly, anti-PD1 and anti-PD-L1 antibodies disrupt PD1-PD-L1 binding, restoring the activation of T-cells and their antitumor activity [10] (Figure 1).

Food and Drug Administration (FDA)-approved agents include inhibitors of CTLA-4 (ipilimumab and tremelimumab), PD-1 (nivolumab, pembrolizumab and cemiplimab), PD-L1 (atezolizumab, avelumab and durvalumab), and lymphocyte-activation gene 3 (LAG-3; relatlimab) [11].

## 4. Immune-Related Adverse Events

The proven efficacy of ICIs against cancer has led to their increased use in oncology, with an increasing number of adverse events reported. The latter are distinct from the toxicity of traditional chemotherapy or molecular targeted therapies [16]. As they promote immune responses, ICIs cause significant immune-related adverse events (irAEs), ranging from inflammation to autoimmunity [17,18].

The frequency and severity of irAEs depends on the type of ICI used [11]. The irAEs that occur with the use of anti-PD-1/PD-L1 drugs are milder and occur at a lower rate than those of anti-CTLA-4 agents [11]. Compared with monotherapies, combined anti-CTLA-4 and anti-PD-1 or anti-PD-L1 therapies increase the risk of irAEs [11,17,19].

A meta-analysis of 23,761 cancer patients treated with ICIs reported an incidence of treatment-related irAEs (any grade) of between 45% and 83% [20]. Severe irAEs (grades 3–5) have been reported in up to 28% of patients [20].

IrAEs typically occur within the first 3–4 months after treatment initiation. However, late onset following prolonged ICI therapy and an association between multi-therapies (anti-CTLA-4 with anti-PD-1 or anti-PD-L1) and earlier onset of irAEs have been described [21].

Table 1 summarises the immune-related adverse events (irAEs) linked to the use of ICIs and their frequency.

## 5. Immune-Related Renal Adverse Events

Renal irAEs are relatively infrequent; however, if they develop and are not treated early, they can lead to critical health issues [22,23].

The most common renal irAE reported in the FDA’s Adverse Events Reporting System is ICI-associated acute kidney injury (AKI) [24,25,26,27,28].

Emerging data show a higher ICI-AKI incidence rate than in initial studies (9.9–29% versus 2–3%) [19,29,30]. This wide range depends, in part, on the different AKI definitions used. In the nephrology field, AKI has generally been reported according to the Kidney Disease Improving Global Outcome (KDIGO) criteria (Table 2), which defines AKI as increases in serum creatine (sCr) compared to baseline, or as diuresis contraction [31]. In contrast, the oncology field reports renal adverse events according to the National Cancer Institute’s Common Terminology Criteria for Adverse Events (NCI-CTCAE; Table 3) which define AKI as sCR increases compared to the “upper limit of normal” (ULN) in the old edition; in the latest version, the NCI-CTCAE does not consider early AKI stages and reserves the most severe grade for when replacement therapy and hospitalisation are needed [32,33]. Therefore, the NCI-CTCAE leads to an underestimation of AKI and does not identify the mildest cases. Alternatively, the incidence of heterogeneity could also depend on different patient characteristics, the different ICIs used, and different tumour types.

To reconcile these different definitions, Gupta et al. have proposed a classification of ICI-AKI by subdividing it into “defined”, “probable”, and “possible” [32].

After the exclusion of other causes and the review of risk factors, they propose to classify an ICI-AKI as “defined” when the diagnosis is confirmed by the outcome of the renal biopsy, as “probable” in the presence of at least two of the following: increased sCr ≥ 50% or the need for renal replacement therapy (RRT) and sterile pyuria (500 white blood cells/hpf) or eosinophilia (500 cells per L), and as “possible” in the presence of an increase in sCr ≥ 50% or a need for RRT [32].

The most typical histopathological finding in patients who develop ICI-AKI is acute tubular interstitial nephritis (ATIN) [19,34,35,36]. In a study by Gupta et al., more than 80% (121/151) of patients undergoing renal biopsy presented with ATIN [36]. However, other histological lesions (e.g., pauci-immune vasculitis or thrombotic microangiopathy) may be found in patients undergoing therapy with ICI who develop AKI [37,38,39,40]. These findings make renal biopsy a fundamental point in the management of oncological patients undergoing therapy with ICI that develop renal abnormalities.

Some symptoms—such as fatigue, dysgeusia, and nausea—are not specific to AKI and can be linked to malignancy [41]; increased sCr and sterile pyuria are the only clinical signs in most cases [22,42]. ICI-AKI can be suspected when haematuria, pyuria, and moderate proteinuria are present with increased sCr levels [43]. Simultaneously, extrarenal irAEs (e.g., colitis, thyroiditis, hypophysitis, dermatitis, or rash) are observed in 43% of patients who develop ICI-AKI [44].

Cortazar et al., in a multicentre study (138 patients with ICI-AKI—defined as a 2-fold increase in sCr or a need for dialysis—and 276 controls), reported that most patients had subnephrotic proteinuria (<3 g/24 h) [44].

An increased risk of developing ICI-AKI has been reported for a lower baseline estimated glomerular filtration rate (eGFR), co-administration of other medications (e.g., proton pump inhibitors (PPIs) and non-steroidal anti-inflammatory drugs (NSAIDs)), combined anticancer therapy with multiple ICIs or simultaneous administration of ICIs and nephrotoxic chemotherapeutics agents, previous or concurrent extrarenal irAEs, and hypertensive patients [27,36,44].

The onset of ICI-AKI varies from 3 to 12 months; the median time of irAE occurrence is shorter with combined ICI therapies than with monotherapies [19,36,42,44,45]. The toxicity of CTLA-4 antagonists can start as early as several weeks after treatment initiation (e.g., 6 to 12 weeks after ipilimumab treatment initiation) [22]. IrAEs develop later for PD-1 inhibitors than for CTLA-4 inhibitors (e.g., 20 weeks and 13.5 weeks after nivolumab and pembrolizumab initiation, respectively) [46].

ICI-AKI is not the only renal irAE observed in patients undergoing ICI therapy; glomerular involvement with or without altered renal function has been described [37].

A systematic review of all biopsy-proven glomerular pathology associated with ICIs shows that more than 20% of the patients have pauci-immune vasculitis (26.7%), podocytopathy (24%; minimal change disease (MCD) and focal segmental glomerulosclerosis (FSGS)) [37]. Less common findings were C3 glomerulonephritis (C3GN; 11.1%), AA amyloidosis (8.9%), IgA nephropathy (8.9%), anti-glomerular basal membrane (GBM) disease (6.7%), membranous nephropathy (2.2%) and lupus-like nephritis (2%) [37]. More than three-quarters (81%) of cases of glomerulopathy are new-onset and are associated with anti-PD-1 or anti-PD-L1 [37]. The onset of ICI-related glomerular disease is variable, at a median of 93 days (interquartile range 44–212 days) [37].

Although not frequent, thrombotic microangiopathy (TMA) has been reported in oncological patients treated with ICIs that develop AKI [38,39]. However, concomitant treatments with drugs such as vascular endothelial growth factor (VEGF) inhibitors, or the simple presence of cancer itself (capable of triggering hemolytic uremic syndrome (HUS) and thrombotic thrombocytopenic purpura (TTP)), do not allow us to draw definitive conclusions on the association between ICI-AKI and TMA [38,39].

ICI-related acid–base and electrolyte disorders have also been described [37,47,48,49]. Hyponatraemia is the most frequent electrolyte disturbance [37,49,50] and may be secondary to ICI-related endocrinopathies (hypophysitis, adrenal insufficiency, and thyroiditis) or renal tubular damage [51,52,53]. In the latter case, hyponatraemia is more frequently associated with other acid–base and electrolyte abnormalities [37,49,50]. Hypokalaemia and metabolic acidosis may indicate distal tubular damage [54].

## 6. Mechanisms of Kidney Damage from Immune Checkpoint Inhibitors

Although several pathogenetic hypotheses have been made, limited data do not allow for definitive conclusions, and the mechanisms of renal damage from ICIs remain un-clarified.

One mechanism by which ICIs may induce kidney damage is related to the activation of autoreactive T-cells normally kept dormant by immune checkpoint mechanisms under physiological conditions. According to this hypothesis, renal involvement may be caused by losing tolerance to an intrinsic renal antigen [11,55]. Another mechanism may involve the reactivation of exhausted drug-specific T-cells previously primed by nephritogenic drugs (e.g., PPIs, NSAIDs) [11,55] (Figure 2). Moreover, kidney damage from ICIs may be linked to the expression of PD-1 and PD-L1 in tubular epithelial cells [11,55]. Anti-PD-L1 and anti-PD-1 could bind to these non-immune cells and cause kidney injury [11,55].

Finally, ICIs contribute to an inflammatory environment in the renal tissue, promoting the migration and activation of effector cells in this tissue, the release of local inflammatory cytokines, and subsequent renal damage [11,55].

## 7. Clinical and Instrumental Exams Used in Clinical Practice to Aid in Diagnosis

Considering the various renal effects from ICIs, a complete evaluation of kidney function (serum creatinine, electrolyte changes, acid–base balance, and urinalysis) is warranted before starting ICI treatment and vigilance is needed during therapy.

An increase in sCr, together with sterile pyuria, is commonly detected in ICI-ATIN [11,22,42,43].

At the onset of signs of renal toxicity, it is essential to exclude all other possible causes of renal injuries, including recent exposure to iodinated contrast medium, patient hydration status, changes to in-home therapies, use of nephritogenic drugs (PPIs, NSAIDs), urinary tract infections, and obstructive causes [56,57,58,59,60].

Non-invasive markers for the definite diagnosis of ICI-AKI are not yet available. Sterile pyuria, active urinary sediment, or mild proteinuria can be signs of renal involvement revealed on urinalysis. In addition, urinary eosinophils and increased serum eosinophils are frequently found, although they are not specific to kidney illness [11]. An increase in the urine albumin–creatinine ratio (uACR) could suggest glomerular or tubular damage, and an elevated uACR should indicate the need for a 24 h urine collection and, eventually, a kidney biopsy. A fractional excretion of urea (FeUrea) of less than 35% in patients undergoing diuretic therapy or a fractional excretion of sodium (FeNa) less than 1% indicates that the AKI is not associated with ICI [11]. A retrospective study identified serum C-reactive protein (CRP) and urine retinol binding protein/urine creatinine (uRBP/Cr) levels as biomarkers in the differential diagnosis between AKI-ICI and renal impairment of other causes [61]. Similarly, another study showed that urinary IL-9 and TNF could aid in diagnosis [62]. However, the limited number of patients treated with ICI included in the studies did not allow definitive conclusions to be drawn.

Gallium-67 scintigraphy has been proposed for non-invasive diagnosis because gallium-67 binds to lactoferrin, which leukocytes release inside the renal interstitium; however, its sensitivity and specificity appear to be low [63]. Similarly, reports of increased absorption of 18 F-fluorodesossiglucose in patients with ATIN-ICI in the renal cortex have been described [64]. However, further studies are needed to determine whether positron emission tomography (PET) is a valid diagnostic test for immuno-mediated nephritis.

## 8. Indications for Kidney Biopsy, ICI Discontinuation, and the Start of Immunosuppressive Therapies

The early recognition of and prompt intervention for irAEs are crucial for preserving organ function; however, understanding the underlying cause of the irAEs is fundamental for preventing unnecessary ICI interruption.

Determining the cause of AKIs can be difficult—especially because concomitant therapies with other nephrotoxic agents contributes to the diagnostic challenge.

When ICI-related renal involvement is strongly suspected, the first step in management includes temporarily discontinuing ICI treatment and starting steroid therapies. In non-responders, an alternative immunosuppressive drug such as mycophenolate mofetil (MMF) or infliximab can be used. Pulsed corticosteroids are usually unnecessary; however, they can be used if there is simultaneous involvement of a second organ, or if the patient is deteriorating rapidly.

There is no consensus that slow or rapid corticosteroid tapering is the best treatment and should be adopted as an individualised approach for each case.

Without convincing data on ICI-related kidney disease, ICI should be continued—at least with milder grades of AKI. A renal biopsy should be performed to differentiate between causes unrelated to ICI and ICI-associated causes.

Despite these general recommendations, different scientific societies propose different strategies (Table 4) [56,57,58,59,60].

For AKI grade 1, ASCO, NCCN, and SITC agree that ICI therapy should be discontinued, while ESMO and AIOM advocate continuing it on a temporary basis [56,57,58,59,60].

For AKI grade 2, all guidelines propose a temporary suspension of ICI and administration of corticosteroids. The only exception is the ESMO guidelines, specifying to continue ICI if AKI is not ICI-related [56,57,58,59,60].

ICI suspension will be permanent in case of AKI grade 3–4, with simultaneous initiation of prednisone or an equivalent drug [56,57,58,59,60].

In case of improvement (up to grade 1) for grades 2, 3, and 4, ASCO, ESMO, and AIOM propose weaning from steroid therapy over 4–12 weeks [56,57,58,59,60].

In case of worsening, ASCO, NCCN, ESMO, and AIOM propose the treating of grade 2 as grade 3, and for grades 3 and 4—except in the AIOM guidelines—they propose the consideration of additional or other immunosuppression therapies [56,57,58,59,60].

Concerning renal biopsy, ESMO suggests performing a renal biopsy in the early stages when possible, and NCCN recommends performing it before starting steroids. On the other hand, ASCO and SITC advise against this in cases of a strong suspicion of AKI-related renal damage [56,57,58,59,60].

In contrast with the oncological approach, the diagnostic nephrological approach suggests using a kidney biopsy as an essential tool for managing nephrotoxicity caused by ICIs, with limitations only for contraindicative cases (e.g., single kidney or coagulopathy). This approach helps to avoid unnecessary ICI interruption or needless exposure to immunosuppressive treatments [11].

Moreover, the execution of the renal biopsy allows for the recognition of any immuno-related glomerular lesions that are otherwise not identifiable. In these cases, treatment is mainly based on the use of corticosteroids (from 1 to 2 mg/kg/day) which may be associated, based on clinical and histological severity, with the use of pulsed steroids—or in case of incomplete response, other immunosuppressive drugs (rituximab, cyclophosphamide, mycophenolate mofetil)—according to the standard therapies for the underlying lesion.

The systematic review of Kitchlu et al. shows complete or partial renal responses (14/17) in patients with pauci-immune glomerulonephritis, renal vasculitis, and podo-cytopathy (FSGS and MCD) [37]. Unfortunately, the interruption of ICI led to the advancement of cancer or death in 9/16 patients [37].

## 9. Recovery from ICI-Mediated irAEs

The decision to continue, terminate, or change treatment is a daily problem and depends mainly on the effectiveness of the therapy. In making these decisions, doctors should discuss with patients the advantages and disadvantages of resuming immunotherapy, considering their expectations and quality of life and the potential risk of end-stage renal disease (ESRD).

Current oncology guidelines support the permanent discontinuation of ICI therapies in patients with severe renal toxicity (grades 3–4), except in selected situations [56,57,58,59,60]. It is crucial to emphasise that ICI therapy is often the only treatment available for managing these patients’ neoplasms, and many irAEs resolve with temporary drug withdrawal, treatment with corticosteroids, and supportive care [11]. Based on these considerations, evaluating ICI rechallenge in cancer patients who previously responded is essential. This concept is supported by a systematic review and meta-analysis performed by Qing Zhao et al. on 789 ICI recovery cases compared with their first ICI treatment. Their data show a similar incidence of severe grade irAEs (*p* > 0.05), despite a higher risk of all-grade irAEs (OR of 3.81; *p* < 0.0001) [63]. Moreover, Cortazar et al. showed a low rate (23%) of recurrence of acute kidney damage following the resumption of ICI therapy [44]. In addition, 80% (8/10) of patients with ICI-induced relapsing AKI show lasting complete or partial remission after receiving infliximab [44].

Many oncologists choose to continue prednisone at low doses (5–10 mg/day) when attempting an ICI rechallenge; however, there is currently no data to support such use [11].

Patients with a history of renal irAEs should be carefully monitored when resuming immunotherapy for early identification of a new event [11]. After a second renal adverse event, definitive discontinuation would be preferred.

## 10. ICIs in Patients with Chronic Kidney Disease and Renal Replacement Therapy

Patients with severe chronic kidney disease (CKD) or renal replacement therapy (dialysis or kidney transplantation) are generally not enrolled in clinical trials, and clinical data are limited [65].

Low baseline eGFR is a risk factor for ICI-AKI [27].

Clinical cases and a series of ESRD patients on dialysis show a frequency of irAEs and an ICI efficacy not significantly different from the general population—probably because ICI clearance during dialysis is minimal; however, additional studies are needed [11,65,66,67,68].

Not surprisingly, transplant patients have high allograft rejection and mortality rates after ICI initiation. A review by Perazella et al., which included 21 patients, showed an acute rejection frequency of 43% [19]. Other studies have suggested that acute rejection is particularly prevalent with combined treatment with multiple ICIs [65,69,70]. In contrast, transplant patients undergoing mTOR inhibitor therapy have the lowest risk of graft rejection without a significant additional risk of cancer progression [65]. ICI-related allograft rejection is generally caused by acute T-cell rejection after prolonged ICI therapy [65]. Prospective studies are needed to optimise immunosuppression and cancer treatment in these patients.

## 11. ICIs in Patients with Pre-existent Autoimmune Glomerulonephritis

Concerns about the risk of reactivation of the underlying disease have led to the exclusion of cancer patients with pre-existing autoimmune diseases from clinical trials on ICIs. The available data are, therefore, poor and mostly concern rheumatological diseases—although reactivations of glomerulonephritis have been reported [71,72,73,74]. However, from the available data, it is difficult to determine the incidence or frequency of exacerbations, but it seems that these are not rare and are more frequent with anti-PD-1/PD-L1 than with anti-CTLA-4 [71,72,73,74].

The data are also poor for deducing whether disease in remission or maintenance therapies can have a protective role [71,72,73,74].

Therefore, oncological patients with pre-existent autoimmune disease must receive a multidisciplinary evaluation before therapy with ICIs, as well as close monitoring during treatment.

In the case of reactivation, autoimmunity therapy should try to affect the goals of cancer treatment as little as possible. For this purpose, corticosteroids have proven effective and have allowed the continuation of therapy with ICIs. Among other immunosuppressors, rituximab—which is the treatment that least suppresses the function of T-cells—would be preferable for its lack of interference with ICIs [71,72,73,74].

## 12. Suggested Nephrology–Oncology Approach

In managing an oncological patient, a multidisciplinary approach is increasingly necessary—taking care of the patient in every phase of the disease, improving the response to oncological treatments and favouring supportive therapies [75]. In this context, the nephrologist is crucial for properly managing renal toxicity from anticancer drugs [75]. A close collaboration between oncologists and nephrologists increases the knowledge of drug-mediated toxicity [75]. In addition, nephrologists should be part of the team when developing guidelines on the management of anticancer therapies—encouraging nephrological points of view and emphasising the importance of specific procedures such as renal biopsy that, although invasive, are often needed to provide the patient with the most appropriate personalised care.

Based on the current guidelines, we recommend assessing renal function in all patients before therapy, with regular follow-up during treatment [56,57,58,59,60]. Correcting predisposing factors could reduce the baseline risk of immune-related adverse reactions. For this purpose, we recommend compensating and self-monitoring blood pressure, suspending the unnecessary use of chronic nephrotoxic drugs, and discouraging the occasional use of nephrotoxic drugs. Follow-up should be at least monthly, and more frequently with evidence of deteriorating renal function or other renal changes (e.g., sterile pyuria, haematuria, proteinuria, altered pH or electrolyte balance). In accordance with oncological guidelines, we suggest excluding the most common causes of renal dysfunction (e.g., dehydration, obstruction, and non-ICI-related nephrotoxicity) before considering possible correlations with ICIs [56,57,58,59,60].

There is little data available for CKD patients undergoing conservative therapies. The increased risk of ICI-AKI for these patients indicates a need for close monitoring, with urine and blood tests at least every two weeks [11].

Considering the actions of immune-system stimuli in patients with a history of autoimmune disease, we recommend starting ICI treatment when the autoimmune disease is in remission and when undergoing maintenance therapy, with follow-up at least every 15 days. For the same reason and due to the increased risk of organ rejection, we recommend starting ICI therapy in renal transplant patients only if it is the best or only therapy available, with weekly follow-up [19].

For patients with no history of kidney damage, we recommend urinalysis and an assessment of creatinine levels before the start of therapy and then monthly—preferably before the next ICI cycle.

The flow chart in Figure 3 shows the suggested approach for the initial evaluation of cancer patients eligible for ICI therapy.

In Figure 4, we present a revised approach informed by a review of different studies in the literature, considering the current guidelines of oncological societies but not neglecting the nephrological point of view [11,41,49,56,57,58,59,60,76,77,78,79].

Although not specific, sterile pyuria or active urinary sediment should increase the suspicion of ICI-related ATIN [11,22,42,43]. Therefore, in these cases, suspension of ICI therapy is preferred even in cases of mild AKI, and immunosuppressive therapy should be started if kidney function does not improve after the suspension of ICI therapy. In detecting proteinuria in the nephrotic range, the suspicion of a pathology unrelated to ICI treatment should be strong; thus, a renal biopsy would be preferable over any other intervention [43,44].

In cases of low-grade AKI, discontinuation of ICI therapy and initiation of corticosteroid therapy may be delayed until histological diagnosis in the absence of a strong suspicion of ICI-AKI [11]. Conversely, the suspension of ICI treatment must be immediate in cases of severe AKI, and the initiation of steroid therapies can only be delayed in cases of rapid renal biopsy (within 48 h) [11]. Cancer patients are at increased risk of ischaemic or nephrotoxic AKI, so it is difficult to distinguish between ICI-related ATIN and other causes.

Therefore, a kidney biopsy will help define the type of kidney injury in these patients. For non-immune related damage, the lesion should be treated, and ICI treatment should resume when kidney function returns to baseline. However, in the case of ICI immuno-related disease, corticosteroid therapy should be initiated and continued for at least three months, followed by a slow reduction [11,56,60]. If the kidney disease is non-responsive, treatment with other immunosuppressants should be evaluated [11,56,60]. The re-introduction of ICIs—discouraged in cases of severe AKI—should depend on several factors (e.g., other possible oncological therapies, disease activity, life expectancy, tumour, and patient’s will). When reintroducing ICIs, contemporary low-dose corticosteroids and changing the type of ICI used should be considered [11].

This recommended approach aims to avoid the unnecessary suspension of ICI therapies and unnecessary corticosteroid treatments for patients lacking other therapeutic options or for whom the risk of renal impairment is not the worst possible alternative.

## Figures and Tables

**Figure 1 cancers-15-01891-f001:**
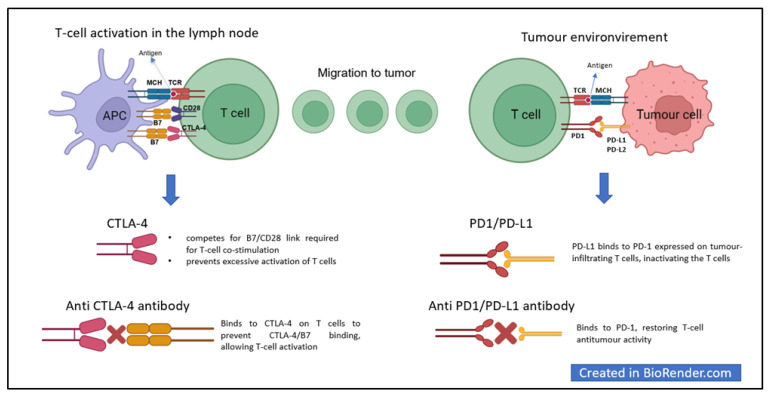
Immune checkpoint pathways and immune checkpoint inhibitors. Legend: APC, antigen-presenting cell; B7, B7 molecules; CD28, cluster of differentiation 28; CTLA-4, cytotoxic T-lymphocyte antigen-4; MHC, major histocompatibility complex; PD-1, programmed death protein 1; PD-L1, programmed death ligand 1; PD-L2, programmed death ligand 2; TCR, T-cell receptor.

**Figure 2 cancers-15-01891-f002:**
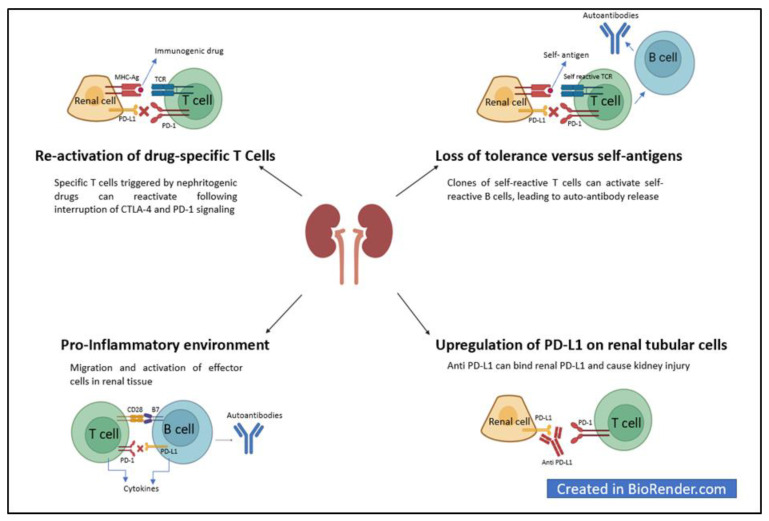
Mechanisms of kidney damage from ICLs. Legend: APC, antigen-presenting cell; B7, B7 molecules; CD28, cluster of differentiation 28; CTLA-4, cytotoxic T-lymphocyte antigen-4; MHC, major histocompatibility complex; PD-1, programmed death protein 1; PD-L1, programmed death ligand 1; PD-L2, programmed death ligand 2; TCR, T-cell receptor.

**Figure 3 cancers-15-01891-f003:**
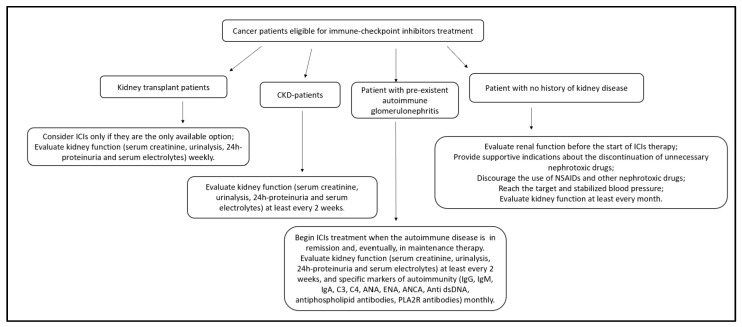
Suggested approach in the initial evaluation of the cancer patients eligible for immune checkpoint treatments. Legend: CKD, chronic kidney disease; ICIs, immune checkpoint inhibitors; IgG, immunoglobulin G; IgM, immunoglobulin M; IgA, immunoglobulin A; C3, complement C3; C4, complement C4; ANA, antinuclear antibody; ENA, extractable nuclear antigens antibodies; ANCA, anti-neutrophil cytoplasmic antibody; anti dsDNA, anti-double-stranded DNA; PLA2R, phospholipase A2 receptors; NSAIDs, non-steroidal anti-inflammatory drugs.

**Figure 4 cancers-15-01891-f004:**
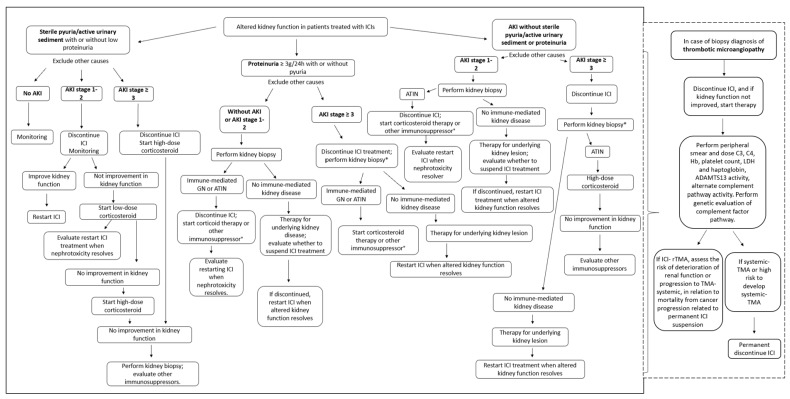
ICI-AKI management. Legend: *, within 48 h, if not possible, start corticosteroid therapy; °, if there is no improvement in kidney function; AKI, acute kidney injury; ATIN, acute tubular-interstitial nephritis; GN, glomerulonephritis; ICI immune checkpoint inhibitor; C3, complement C3; C4, complement C4; Hb, hemoglobin; LDH, lactate dehydrogenase; ICI- rTMA immune checkpoint inhibitor- renal thrombotic microangiopathy.

**Table 1 cancers-15-01891-t001:** Summary of the immune-related adverse events (irAEs) linked to the use of ICIs.

Type of irAEs	Any Grade Toxicity (% of Patients)	Grade 3–4 Toxicity (% of Patients)
SKIN (rash, pruritus, psoriasis, vitiligo, DRESS, Stevens-Johnson syndrome)	13–50	<3
GASTROINTESTINAL (diarrhoea, colitis, ileitis, pancreatitis)	16–54	1–11
LIVER (hepatitis)	5–10	1–2
ENDOCRINE (hyper or hypothyroidism, hypophysitis, adrenal insufficiency, diabetes)	5–21	0
RESPIRATORY (pneumonitis, pleuritis, sarcoid-like granulomatosis)	20–30	1–9
CARDIOVASCULAR (myocarditis, pericarditis, vasculitis)	<1	0
NEUROLOGIC (Neuropathy, Guillain-Barrè syndrome, myelopathy, meningitis, encephalitis, myasthenia)	1–4	0
EYE (uveitis, conjunctivitis, scleritis, episcleritis, blepharitis, retinitis)	<1	0
RENAL (ATIN, glomerulonephritis, tubular acidosis, electrolytes alterations)	1–29	2
BLOOD (haemolyticanaemia, thrombocytopenia, neutropenia, haemophilia)	<1	0
RHEUMATIC (polymyalgia rheumatica, psoriatic arthritis, seronegative-polyarthritis, dermatomyositis, myositis)	2–12	<1 (myositis)

**Table 2 cancers-15-01891-t002:** KDIGO criteria for AKI.

KDIGO		
Stage	Serum Creatinine	Diuresis
1	1.5–1.9 × baseline or increase ≥ 0.3 mg/dL within 48 h	<0.5 mL/kg/h for 6–12 h
2	2–2.9 × baseline	<0.5 mL/kg/h for >12 h
3	3 × baseline or increase ≥ 4 mg/dL within 48 h or initiation of RRT or in patients < 18 years old, decreased eGFR < 35 mL/min/1.73 m^2^	<0.3 mL/kg/h for >24 h or anuria for ≥ 12 h

**Table 3 cancers-15-01891-t003:** CTCAE criteria for AKI.

CTCAE 3.0		CTCAE 5.0
Grade	Serum Creatinine	Indications
1	1–1.5 × ULN	-
2	1.5–3 × ULN	-
3	>3 × ULN	Hospitalisation
4	>6 × ULN	Dialysis

Legend: ULN, the upper limit of normal.

**Table 4 cancers-15-01891-t004:** Management of AKI in patients undergoing ICI therapies, according to national and international guidelines. Legend: ASCO, American Society of Clinical Oncology; NCCN, National Comprehensive Cancer Network; SITC, Society for Immunotherapy of Cancer; ESMO, European Society for Medical Oncology; AIOM, Italian Association of Medical Oncology.

General Management			Supportive Care; Withdraw Nephrotoxic Medication; Evaluate Other Causes		
			Grade 1	Grade 2	Grade 3–4
ASCO [56]	ICI therapy		Consider temporarily withholding	Temporarily withhold.	If there is a strong suspicion of AKI-ICI, permanently discontinue
	Treatment		Follow-up	Start prednisone or equivalent 0.5–1 mg/kg/d	Start 1–2 mg/kg/d prednisone or equivalent
	Response	Improvement (to grade 1)	Follow-up	Wean CS over 4 weeks	
		Worsening	Treat as grade 2	Treat as grade 3	Consider additional immunosuppressors (such as MMF ° or infliximab *)
	Nephrological consul			Consult nephrologist	
	Kidney biopsy		Kidney biopsy should be discouraged with strong suspicions of ICI-related renal damage until steroid treatment has been attempted		
NCCN [57]	ICI therapy		Consider temporarily withholding	Withhold ICI	Withhold ICI
	Treatment		Follow-up	Start prednisone 0.5–1 mg/kg/d	Start prednisone/methylprednisolone 1–2 mg/kg/d Consider hospitalization
	Response	Improvement (to grade 1)	Follow-up		
		Worsening		Prednisone/methylprednisolone 1–2 mg/kg/d	If kidney injury remains > G2 after 4–6 weeks of steroids, consider other immunosuppressors (such as MMF ° or infliximab *)
	Nephrological consul		Consult nephrologist if not improved within 2 weeks	Consult nephrologist	
	Kidney biopsy			Consider renal biopsy, if feasible, prior to starting steroids	
SITC [58]	ICI therapy		Consider temporarily withholding ICI	Withhold ICI	
	Treatment			Start steroid therapy	
	Response	Improvement (to grade 1)			
		Worsening		Consider other immunosuppressors (such as MMF ° or infliximab *)	
	Nephrological consul		For progressive or persistent AKI grade 1	Consult nephrologist	
	Kidney biopsy			In the suspicion of renal damage that is not ICI-related	
ESMO [59]	ICI therapy		Continue ICI	Continue ICI if not attributed to an irAE.	Withhold ICI
	Treatment			Start 0.5–1 mg/kg/d prednisolone if attributed to an irAE.	
	Response	Improvement (to grade 1)		Wean CS over 4 weeks	Wean corticosteroid over 4–12 weeks
		Worsening	Treat as grade 2	Treat as grade 3	Start prednisolone 1 mg/kg/d or pulse dose methylprednisolone 250–500 mg for 3 days
	Nephrological consult			Consult nephrologist	
	Kidney biopsy			Early consideration of renal biopsy	
AIOM [60]	ICI therapy		Continue ICI	Discontinue ICI	Permanent discontinue ICI
	Treatment			Prednisone 0.5–1 mg/kg/d	Prednisone 1–2 mg/kg/d
	Response	Improvement (to grade 1)		Wean corticosteroid over at least 4 weeks	
		Worsening		Start 1–2 mg/kg/d prednisone	Additional immunosuppressor not indicated
	Nephrological consult		Not indicated		
	Kidney biopsy		Not indicated		

° 1 g × 2/die, weaned over a period of at least 2 months; * 5 mg/kg, maximum 1–3 dose.

## Data Availability

No new data were created.

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
