# Peer review of "Immune Checkpoint Inhibitors and the Kidney: A Focus on Diagnosis and Management for Personalised Medicine"

_cancers, 2023, doi:10.3390/cancers15061891_

Round 1
Reviewer 1 Report
This is an interesting narrative review of Renal irAEs induced by ICI
The authors described several aspects of ICI toxicity including pathogenesis, clinical manifestations, and treatment recomendations including a clinical approach. The authors also included some illustrative figures for the mechanisms of actions of CPI as well as the pathogenesis of renal ICI toxicity.
This review could be of interest to CANCERS readers including not only Oncologists but also some others (Nephrologist, Internal medicine specialists)
Some aspects should be clarified before acceptance, including:
Major aspects
Please check table 1 Rheumatologic side effects. Expand for other more common irAEs, including PMR-Like, Psoriatic arthritis-Like, and myositis. Additionally, on some occasions, specially myositis is grades 3-4 (correct percentage in table)
In section 8. Please differentiate the treatment of AKI and the treatment of immune-related GMN, as well as, infliximab and MMF doses and duration.
The article included 5 heavy tables about recommendations from different societies (NCNN, ASCO, SITC, ESMO, AIOM). Could you briefly describe the differences and similarities among those recommendations and try to collapse the information maybe in 1 or 2 tables?
Just a brief comment is made for patients with a history of Autoimmune disease, but a not specific description is made for some GMNs. Is the same for an IgA nephropathy than for instance ANCA-associated vasculitis or SLE?
Additionally in Figure 4, please add in the square of recommendations for patients with Autoimmune GMN some other components such as autoantibodies (i.e ANCAs, Anti dsDNA, antiphospholipid antibodies, PLA2R) or complement levels
Is there any evidence of biomarkers for the diagnosis and monitoring of patients with ICI renal toxicity? NGAL, MCP-1, urinary eosinophils, etc?
Finally, the authors did not include the possibility of kidney thrombotic microangiopathy. This is an important issue for patients with an oncologic underlying disease. A brief statement is recommended as well as an additional square in Figure 5.
Reviewer 2 Report
Excellent work:
Some Minor comments:
- Please explain in detail about histopathology findings other than ATIN
-Please add definition of definite/probable/possible ICPI-AKI.
-Role of imaging ?
